# LightWM: Training-Free Hierarchical Working Memory for Small Language Model Agents

Ziyi Wang[1]  Haonan Jin[1]  Zian Wang[1]  Wendong Wang[1]  Lanshan Zhang[1]

## Abstract

Small language models (SLMs) are attractive for low-cost agent deployment, and their growing capabilities make them promising for procedure-centric workloads that repeatedly execute specialized task families with similar workflows. However, SLM agents still struggle in long-horizon interaction because limited context budgets make it difficult to reliably retain and reuse decision-relevant state across many steps. Existing working-memory methods attempt to mitigate context growth, but their reliance on unstructured natural-language summarization can discard critical facts, introduce state drift, and compound errors in SLM execution. We present LightWM, a training-free hierarchical working-memory framework that decomposes procedure-centric tasks into subgoals and organizes memory into task-level global memory and subtask-level local memory, where local memory directly conditions SLM action selection and is updated from new observations through structured updates. To instantiate such memories without training, a one-time offline LLM-based induction pipeline builds reusable schemas per task family from a few successful traces, requiring no SLM parameter updates or online LLM calls. On ALFWorld `valid_unseen`, Qwen3-4B reaches 0.910 success, whereas representative prompting and prior working-memory baselines under the same setting remain below 0.320.

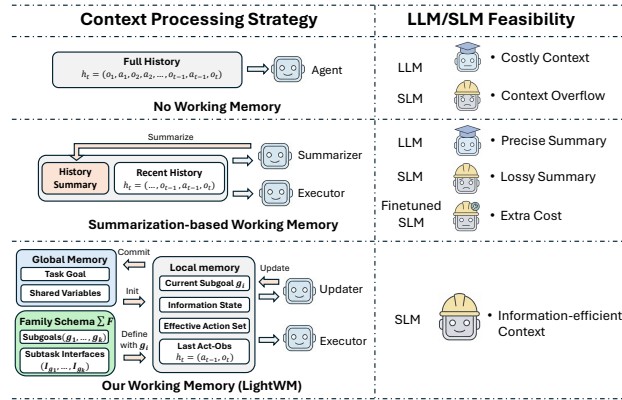

*Figure 1.* A comparison of context-processing strategies for long-horizon agent tasks: using raw history, summarization-based working memory, and our hierarchical goal-conditioned working memory. The figure highlights how limited context budgets make SLMs brittle to long histories and lossy summaries, while structured global/local memory provides a more information-efficient context.

## 1. Introduction

In recent years, Large Language Models (LLMs) have demonstrated remarkable capability in long-horizon, multi-step interactive agent tasks, enabling complex reasoning, planning, and tool use (Yao et al., 2023b; Wang et al., 2023a; Xu et al., 2023; Schick et al., 2023; Patil et al., 2024). However, deploying LLM-based agents in practice still faces substantial constraints, including high inference cost, high latency, and reliance on high-end compute, which makes them less suitable for on-device deployment and high-concurrency settings (Erdogan et al., 2024; Abhyankar et al., 2024).

Against this backdrop, Small Language Models (SLMs) with up to 8B parameters have emerged as a promising alternative for agent applications due to their lower inference cost and improved deployability (Belcak et al., 2025; Erdogan et al., 2024). These advantages are especially relevant for procedure-centric workloads, where agents repeatedly execute specialized task families with similar workflows under tight deployment constraints. Although SLMs have made notable progress in instruction following and basic reasoning, naively replacing an LLM with an SLM in agent systems often leads to a pronounced performance drop (Shi et al., 2023). A central bottleneck is contextual information efficiency: long-horizon interaction produces an ever-growing action–observation history, while SLMs have tighter context and weaker implicit state-reconstruction abil-

[1]Beijing University of Posts and Telecommunications, Beijing, China. Correspondence to: Lanshan Zhang <zls326@sina.com>.

ity than larger models (Jiang et al., 2024; Cai et al., 2024; Tang et al., 2024; Dong et al., 2024; Jin et al., 2024).

To improve contextual information efficiency, prior work—often inspired by the notion of working memory in cognitive science—has explored agent working memory (Hu et al., 2025b; Packer et al., 2024; Hu et al., 2025a), aiming to turn the context window from a passive buffer into a controllable, updatable, and interference-resistant workspace, thereby increasing the utility of information within the context window. Most existing working-memory methods represent memory in unstructured or weakly structured natural language and update it via summarization (Packer et al., 2024; Hu et al., 2025a). For SLMs, this design can be brittle: compression may omit critical facts or introduce state drift, and such errors can accumulate over long interactions under tight context budgets.

Some studies attempt to address this by fine-tuning or distilling small-to-medium LLMs (e.g., 8–30B models) so that they learn to construct and use working memory more effectively, achieving measurable gains (Mu et al., 2023; Ge et al., 2024; Liao et al., 2025; Wang et al., 2024; Yu et al., 2026; Zhou et al., 2026). In realistic agent deployments, however, these learned approaches may require additional post-training and task-specific data construction, making them costly to adapt when task families or deployment conditions change.

These limitations raise a key question: **For procedure-centric long-horizon tasks, before resorting to costly post-training, can we design a working memory that SLMs can use reliably—by making memory more information-dense and decision-relevant—so as to reduce dependence on large-scale data construction and fine-tuning?**

From a Partially Observable Markov Decision Process (POMDP) perspective, effective decision making does not require the full action–observation history, but only information that is necessary for selecting the next action (Kaelbling et al., 1998; Bertsekas & Shreve, 1996). In agent tasks, such decision-relevant information typically includes: (1) the current goal or subgoal, which determines what outcomes are prioritized; (2) the information state inferred from observations that is relevant to assessing progress; and (3) the action set that is effective currently. While LLMs can often recover these factors implicitly from verbose natural-language context, SLMs are less reliable at doing so, motivating a working-memory design that makes these elements explicit.

Motivated by the analysis above, we propose LightWM, a training-free hierarchical working-memory framework tailored to SLM-based agents. To mitigate forgetting in long-horizon tasks and to reduce context-window pressure, our framework decomposes a task into multiple subtasks from the perspective of goal, and organizes working memory into two layers: an outer global memory at the task level and an inner local memory at the subtask level. Local memory manages information specific to the current subgoal and interfaces directly with the prompt used by the SLM. The SLM is used as a state updater to update local memory based on new observations, and as an executor to generate actions conditioned on the local memory. Global memory maintains information relevant to the overall task goal and serves as a bridge for information transfer across subtasks by managing the inputs and outputs of local memory.

Within this hierarchical working-memory design, the content of working memory is explicitly goal-conditioned: what constitutes the relevant information state and the effective action set depends on the active (sub)goal. Concretely, we represent local memory with a structured schema whose fields specify (1) the minimal set of observations and inferred facts needed to assess progress toward the current subgoal, and (2) the currently feasible actions together with their applicability conditions under the current subgoal. This schema-based representation suppresses subgoal-irrelevant details and yields a denser, more decision-relevant context for SLM execution and memory updating.

To instantiate such memories without post-training the deployed SLM, we use a one-time offline LLM-based induction pipeline that builds a reusable schema $\Sigma_{\mathcal{F}}$ from a few successful traces for each task family. Here, a task family denotes tasks sharing a high-level template, completion criteria, and interaction procedure. The induced schema defines how hierarchical memory is organized, updated, and exposed to the SLM during execution, and is reused only across instances within the same family rather than assumed to be universal. During online execution, the deployed SLM runs under this fixed schema without additional LLM calls or parameter updates, reducing reliance on task-specific post-training or repeated model adaptation.

We evaluate LightWM on the ALFWorld valid unseen split with 145 tasks across six task families, using task success rate as the end-to-end metric (Shridhar et al., 2021). To reflect deployment constraints for small agents, we instantiate the same 4B backbone under two inference precisions, FP8 and FP16. On this challenging split, existing prompting and working-memory baselines remain brittle for SLM agents, whereas our goal-conditioned, schema-based memory enables Qwen3-4B to reach a 0.910 success rate. We further include Gemini-2.5 Flash as a stronger reference model under the same protocol, where our memory improves over a direct ReAct agent from 0.786 to 0.945 success (Yao et al., 2023b). Together, these results support our central claim that explicit, goal-conditioned working memory can substantially improve structured long-horizon agent execution, especially for SLMs under tight context and deployment constraints.

**Conflict of Interest Disclosure.** The authors declare no financial conflicts of interest related to this work.

## 2. Preliminaries

**Agentic Interactive Tasks and POMDPs.** We study long-horizon interactive agent tasks in which an agent repeatedly observes an environment and issues actions to accomplish a specified objective. We model the environment as a POMDP $\mathcal{M} = (\mathcal{S}, \mathcal{A}, \mathcal{O}, T, Z)$, where $s_t \in \mathcal{S}$ is the latent state, $a_t \in \mathcal{A}$ the executed action, and $o_t \in \mathcal{O}$ the agent's observation. The environment transitions as $s_{t+1} \sim T(\cdot \mid s_t, a_t)$ and emits observations as $o_{t+1} \sim Z(\cdot \mid s_{t+1}, a_t)$. Due to partial observability, decisions condition on the interaction history $h_t = (o_1, a_1, o_2, a_2, \ldots, a_{t-1}, o_t)$. We evaluate performance by task completion (success), rather than stepwise reward optimization.

**Decision-Relevant Information State.** In POMDPs, optimal control can be based on an *information state*, i.e., any sufficient statistic of $h_t$ for decision making; the belief state is a canonical example (Kaelbling et al., 1998; Bertsekas & Shreve, 1996). In language-grounded environments, explicitly maintaining an exact belief (or any exact sufficient statistic) is often impractical due to high-dimensional observations, large latent state spaces, and the need for an explicit or learned transition/observation model. We therefore use a weaker abstraction: a *decision-relevant information state* $m_t = f(h_t)$ that approximately preserves goal-conditioned facts needed for reliable action selection, without being strictly sufficient.

**Goals, Subgoals, and Subtask Types.** Each task instance specifies a high-level goal $G$ and can be decomposed into subgoals $\{g_k\}_{k=1}^K$. We focus on *procedure-centric* task settings, where instances within a family share a reusable high-level workflow, observable subtask-completion signals, and parameterized action patterns, while differing in entities and initial conditions. We further group subtasks into *subtask types* that share similar completion criteria and interaction procedures; this will support reusable, type-specific memory and action specifications.

**Environment Actions and Effective Action Set.** We distinguish the environment action set $\mathcal{A}$ from an agent-centric effective action set, which captures action patterns that are likely to make progress toward the subgoal under the current context. In language-based settings, the agent generates a token sequence that is parsed into an environment action, so formatting or parsing errors can yield invalid actions.

**Type-Conditioned Action Specification via Pruned Grammar.** Many environments provide an action grammar or a table of templates defining parseable command

*Table 1.* Notation used in Method.

| SYMBOL | MEANING |
| --- | --- |
| $G$ | TASK GOAL (NATURAL LANGUAGE) |
| $\mathcal{F}$ | TASK FAMILY (SHARED TEMPLATE + SUBTASK PATTERN) |
| $(g_1, \ldots, g_K)$ | FIXED SUBTASK SEQUENCE FOR $\mathcal{F}$ |
| $I_{g_k}$ | SUBTASK INTERFACE TEMPLATE FOR SUBTASK $g_k$ |
| $M_t^G$ | GLOBAL MEMORY AT STEP $t$ |
| $M_{t,k}^L$ | LOCAL MEMORY FOR SUBTASK $g_k$ AT STEP $t$ |
| $\Sigma_{\mathcal{F}}$ | INDUCED SCHEMA FOR FAMILY $\mathcal{F}$ |
| $\pi_{\text{UPD}}$ | STATE UPDATER (SLM) PRODUCING PATCH OPERATIONS $\Delta_t$ |
| $\pi_{\text{EXE}}$ | EXECUTOR (SLM) PRODUCING ACTION $a_t$ |
| $\Delta_t$ | STRUCTURED PATCH OPERATIONS APPLIED TO LOCAL MEMORY |

forms. For each subtask type, we derive a compact, goal-conditioned action specification by pruning the full grammar to retain action types typically required for successful execution. This specification is exposed as a prompt-side action template that guides the model toward the progress-making region. Importantly, it is an approximation to the underlying effective action set rather than an exact characterization.

**Working Memory and Hierarchy.** We model working memory as an explicit, externally maintained state used for action generation and updated after each new observation. To capture subgoal structure, we maintain a task-level *global* memory $M_t^G$ and a subgoal-level *local* memory $M_t^L$, where local memory stores subgoal-specific decision-relevant information and the type-conditioned action specification. Local memory is initialized from global memory at subgoal switches and selectively committed back upon subgoal completion, enabling information transfer across subtasks under tight context budgets.

## 3. Method

### 3.1. Problem Setup and Objective

We consider long-horizon interactive tasks where an agent repeatedly receives observations $o_t$, generates actions $a_t$, and interacts with the environment to accomplish a natural-language goal $G$. Task success is determined by the environment at episode end.

We focus on a *deployment-oriented* setting: enabling small language models (SLMs) to execute long-horizon tasks under tight context without any parameter updates. The key challenge is to expose to the SLM only the decision-relevant information needed at each execution step, without relying on an ever-growing natural-language history.

Motivated by the POMDP view that control can be based

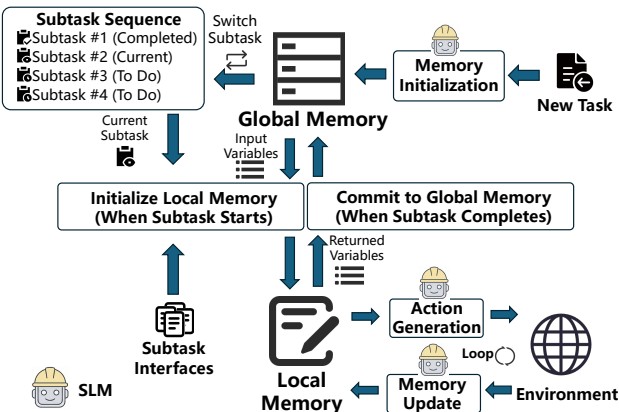

*Figure 2.* Online execution workflow with schema-governed hierarchical working memory. A new task initializes global memory and a subtask sequence. At each subtask boundary, task-level input variables and the current subtask interface instantiate local memory. The SLM alternates between memory update and action generation while interacting with the environment. Once a subtask completes, returned variables are committed back to global memory, and the controller switches to the next subtask.

on a compact information state, we construct a goal-conditioned hierarchical working memory that specifies (i) what information is stored, (ii) how it is updated, and (iii) what is exposed for action generation.

## 3.2. Notation and Task Families

We index interaction steps by $t = 1, \ldots, T$. A *task family* $\mathcal{F}$ is a set of task instances that share the same goal template and execution pattern, while differing in entity instantiations and initial conditions. Each family is associated with a fixed subtask sequence $(g_1, \ldots, g_K)$ that serves as an *execution skeleton*: it defines subtask boundaries and information flow but does not prescribe instance-level strategies.

We maintain a task-level *global memory* $M_t^G$ that persists across subtasks, and a subtask-level *local memory* $M_{t,k}^L$ instantiated for the active subtask $g_k$. A family-specific schema $\Sigma_F$ specifies the subtask sequence $(g_1, \ldots, g_K)$ and a set of per-subtask interfaces $(I_{g_1}, \ldots, I_{g_K})$. Each subtask interface $I_{g_k}$ is a static schema-level template that defines the local-memory fields, input variables, action templates, update constraints, and completion outputs for subtask $g_k$. At runtime, $I_{g_k}$ is instantiated with task-level variables from $M_t^G$ to form the active local memory $M_{t,k}^L$.

## 3.3. Overview: Training-Free Hierarchical Working Memory

Our framework is training-free and consists of two phases. In an **offline** phase performed once per task family, we induce a reusable schema $\Sigma_{\mathcal{F}}$ from a small number of successful trajectories; $\Sigma_{\mathcal{F}}$ defines the subtask sequence, per-

subtask interfaces, action templates, and update constraints. A concrete induction example is provided in Appendix B. In an **online** phase performed per episode, the SLM executes the task under the fixed induced schema, without any additional LLM calls or parameter updates.[1]

Figure 2 illustrates the online execution workflow. A new task initializes global memory and a subtask sequence. At each subtask boundary, task-level input variables and the current subtask schema instantiate the active local memory. During execution, the SLM alternates between local-memory update and action generation while interacting with the environment. Once the subtask completes, returned variables are committed back to global memory, and the controller switches to the next subtask.

## 3.4. Memory Representation

### 3.4.1. GLOBAL MEMORY

Global Memory $M^G$ maintains task-level information shared across subtasks. It stores (i) a structured goal interpretation used to guide subtask execution, (ii) task-relevant symbolic facts accumulated during execution (e.g., identified targets, completed operations), and (iii) execution context variables such as the agent's location, inventory, and extracted interactables. Global memory is updated by (a) facts extracted from observations and (b) structured outputs committed from completed subtasks.

### 3.4.2. LOCAL MEMORY INSTANTIATED FROM A SUBTASK INTERFACE

For each subtask type, $\Sigma_{\mathcal{F}}$ contains a subtask interface $I_g$, represented as a structured template with fixed fields and empty or default-valued writable slots. At subtask start, $I_g$ is instantiated with task-level input variables from $M^G$ to form the active local memory $M^L$. $M^L$ is not a free-form history buffer: it exposes only decision-relevant fields to the executor and constrains how the updater may change state. Concretely, each subtask defines five fields (Table 2). During online execution, the updater produces rule-constrained patches that can modify CONTEXT and RETURN only; all other fields are read-only.

**Patch-based updates.** The updater outputs a list of patch operations (e.g., SET(path, value), LIST_REMOVE(path, value)). Patches violating RULES are rejected, avoiding free-form summarization. For instance, in SEARCH_OBJ, if the agent arrives at receptacle $X$ and observes the target object, the updater must set return.target_object_location and may additionally remove $X$ from the search frontier in the

---

[1]Research code and artifacts are available at https://github.com/BloomChant/LightWM.

*Table 2.* Local memory fields. Only CONTEXT and RETURN are writable online via rule-constrained patches.

| FIELD | ROLE | R/W |
|---|---|---|
| CORE | READ-ONLY SUBTASK IDENTITY, GOAL, AND NATURAL-LANGUAGE TERMINATION CONDITION | R |
| CONTEXT | MINIMAL GOAL-CONDITIONED INFORMATION STATE FOR THE ACTIVE SUBTASK (INITIALIZED FROM $M^G$ AND UPDATED ONLINE) | R/W |
| ACTIONS | SUBTASK-SPECIFIC ACTION TEMPLATES AND EXECUTOR CONSTRAINTS (OFFLINE, FIXED) | R |
| RULES | TYPED CONSTRAINTS FOR ALLOWABLE UPDATES; DEFINES THE PATCH GRAMMAR AND GUARDED OPERATORS | R |
| RETURN | STRUCTURED SUBTASK OUTPUTS COMMITTED BACK TO $M^G$ UPON COMPLETION | R/W |

---

**Algorithm 1** Online execution with hierarchical working memory

---

**Require:** Goal $G$, step cap $T$, schema $\Sigma_{\mathcal{F}}$ containing subtask sequence $(g_1, \ldots, g_K)$ and interfaces $(I_{g_1}, \ldots, I_{g_K})$
1: Initialize $t \leftarrow 1$ and receive initial observation $o_1$
2: $M^G \leftarrow$ INITGLOBAL$(G, o_1; \Sigma_{\mathcal{F}})$
3: **for** $k = 1$ to $K$ **do**
4:      $M^L \leftarrow$ INITLOCAL$(M^G, I_{g_k}; \Sigma_{\mathcal{F}})$
5:      **while** NOTDONE$(M^L.$RETURN$)$ **and** $t \leq T$ **do**
6:          $\Delta_t \leftarrow \pi_{\mathrm{upd}}(o_t, M^L)$
7:          $M^L \leftarrow$ APPLYPATCH$(M^L, \Delta_t, M^L.$RULES$)$
8:          **if** DONE$(M^L.$RETURN$)$ **then**
9:             **break**
10:         **end if**
11:         $a_t \leftarrow \pi_{\mathrm{exe}}(M^L.$CONTEXT$, M^L.$ACTIONS$)$
12:         Execute $a_t$; receive $o_{t+1}$; set $t \leftarrow t + 1$
13:      **end while**
14:      $M^G \leftarrow$ COMMIT$(M^G, M^L.$RETURN$)$
15: **end for**
16: **return** success/failure from the environment

---

same patch.

**Subtask termination.** Each subtask defines required output fields $\mathcal{R}_k$. The subtask terminates when all required fields become non-null: $\forall r \in \mathcal{R}_k$, $M^L.$RETURN$[r] \neq \varnothing$. Upon termination, RETURN is committed to $M^G$ and execution proceeds to the next subtask in the fixed sequence. CORE stores a natural-language termination description for the SLM; the controller deterministically checks termination via required RETURN fields.

### 3.5. Online Execution: Updater and Executor

Online execution alternates between two SLM calls with distinct responsibilities. The *State Updater* $\pi_{\mathrm{upd}}$ consumes the latest observation and the current local memory, and outputs a rule-constrained patch $\Delta_t$ to update CONTEXT and RETURN. The *Executor* $\pi_{\mathrm{exe}}$ reads the updated local memory together with the fixed subtask action templates and generates an environment action $a_t$ (ReAct-style). The overall loop is summarized in Algorithm 1.

### 3.6. Offline Schema Induction (per Task Family)

The offline phase induces a reusable schema $\Sigma_{\mathcal{F}}$ for each task family without parameter updates. Operationally, we (i) elicit the shared subtask sequence and per-subtask requirements (local fields, action templates, and update constraints) from a small number of successful trajectories, and (ii) constructing a family-specific schema by extending and adapting a base schema of a reference family (e.g., *pick-and-place*) using an LLM. The resulting $\Sigma_{\mathcal{F}}$ is reused across instances within the same family, enabling scalable deployment without continuous data construction or model training. The induction is performed under fixed controller-facing representation guidelines, which specify controller-

compatible formatting conventions, field styles, and patch operators. These guidelines constrain the form of the generated schema, while the family-specific subtask sequence, required information-state variables, completion predicates, and action patterns are inferred from successful trajectories.

## 4. Experiments

In this section, we evaluate LightWM on long-horizon embodied instruction following in ALFWorld. We first describe the evaluation protocol, models, and implementation details (Sec. 4.1), then introduce baselines representing representative prompting and prior working-memory mechanisms (Sec. 4.2). We report task success rates across multiple backbones (Sec. 4.3), followed by lightweight yet informative analyses based on the completed runs: step usage and token/latency cost, including a per-step token growth trend and the one-time offline schema induction cost (Sec. 4.5). Finally, we present log-based failure mode statistics such as invalid-action rates and step-cap terminations (Sec. 4.6).

### 4.1. Experimental Setup

We evaluate on ALFWorld `valid_unseen`, which contains 145 tasks across six task families (following the standard split). Each episode is successful if the environment returns a terminal success signal; we report *success rate* over all tasks.

We use Qwen3-4B-Instruct under FP8 and FP16 inference, Qwen3-8B under FP16, and Gemini-2.5 Flash as an API-based reference model. Unless stated otherwise, all methods (including memory updates and summarization) use the same evaluated backbone to ensure a fair comparison. We cap each episode by a maximum of $T = 50$ environ-

*Table 3.* Success rate on ALFWorld `valid_unseen`

| METHOD | QWEN3-4B (FP8) | QWEN3-4B (FP16) | QWEN3-8B (FP16) | GEMINI-2.5 FLASH |
|---|---|---|---|---|
| REACT (YAO ET AL., 2023B) | 0.048 | 0.062 | 0.579 | 0.786 |
| SIMPLY SUMMARIZE | 0.179 | 0.228 | 0.469 | 0.531 |
| HIAGENT (HU ET AL., 2025A) | 0.235 | 0.248 | 0.503 | 0.558 |
| MEMGPT (PACKER ET AL., 2024) | 0.298 | 0.317 | 0.497 | 0.821 |
| LIGHTWM(OURS) | **0.821** | **0.910** | **0.828** | **0.945** |

ment steps (the *step cap*); successful episodes terminate early when the environment signals success, while unsuccessful episodes terminate when reaching the step cap.[2] We use greedy decoding (temperature 0, `top_p = 1`, `n = 1`) together with the environment action grammar.

### 4.2. Baselines

We compare to representative prompting and prior working-memory baselines. **ReAct** (Yao et al., 2023b) conditions the model on the environment-provided action grammar and generates (thought, action) outputs step-by-step. **Simply Summarize** periodically replaces recent interaction history with a model-generated natural-language summary. We use a frequency of every 5 interactions with environment in the main setting. **HiAgent** (Hu et al., 2025a) compresses context at the subgoal granularity; when a subgoal is marked complete, the model summarizes the subgoal's execution trace into a compact representation for subsequent steps. **MemGPT** (Packer et al., 2024) partitions the prompt into a fixed-size writable working context and a FIFO interaction buffer; the buffer is capped at 10 interactions, and when this limit is reached, the earliest 5 interactions are summarized into the working context.

Our method maintains a schema-governed hierarchical working memory induced offline once per task family and reused across task instances. Online execution updates structured fields via rule-constrained patch operations and conditions action generation on subtask-specific action templates.

### 4.3. Main Results

Table 3 reports success rates on ALFWorld dataset split `valid_unseen`. Overall, baseline prompting and prior working-memory methods remain brittle for small backbones in long-horizon settings, while our goal-conditioned, schema-governed working memory substantially improves end-to-end success. We also observe consistent improvements on a stronger API model, suggesting that making decision-relevant state and feasible actions explicit can ben-

efit both SLMs and LLMs.

### 4.4. Component Ablation and Robustness

We next examine the contribution of each component in our framework and assess whether the reported performance is robust beyond deterministic decoding. We decompose our method into three components: hierarchical goal-conditioned working memory (HWM), subtask-specific action templates (Act.), and rule-constrained patch updates (Patch). HWM maintains the global/local memory hierarchy and exposes a compact subtask-level information state to the executor; Act. provides the subtask-specific action specification; Patch constrains online memory updates to valid operations over writable fields. The ablation is conducted on all 145 ALFWorld `valid_unseen` tasks using Qwen3-4B under FP16 inference.

Table 4 shows that the primary performance gain does not come from restricting the action space. Removing action templates reduces success from 0.910 to 0.821, indicating that Act. improves reliability but accounts for only a bounded gain. Removing patch constraints reduces success to 0.766 and substantially increases output length, suggesting that Patch helps avoid verbose and error-prone full-memory rewrites. Most importantly, the HWM-only variant, which removes both action templates and patch constraints, still achieves 0.697 success, more than double MemGPT's 0.317 under the same backbone. These results indicate that explicit hierarchical, goal-conditioned working memory is the main driver of the improvement, while action templates and patch constraints provide additional reliability.

All main experiments use greedy decoding with temperature 0, top-$p = 1$, and $n = 1$ to ensure reproducibility under a fixed protocol. To further assess robustness under stochastic decoding, we run our full system with temperature 0.7 over five random seeds. The resulting success rates are 0.937, 0.931, 0.917, 0.952, and 0.924, respectively. The consistently high success rates and low variation indicate that the performance gain is not a single-run artifact of deterministic decoding.

---

[2]We use the term *step cap* to denote the maximum number of environment actions allowed per episode; it is a standard safeguard for long-horizon interactive evaluation. We also apply lightweight runtime guardrails to prevent degenerate loops (Sec. 4.5); such early stopping is rare in our method.

*Table 4.* Component ablation on ALFWorld `valid_unseen` with Qwen3-4B (FP16). HWM denotes the hierarchical goal-conditioned working-memory representation; Act. denotes subtask-specific action templates; Patch denotes rule-constrained patch updates. Tok/call reports average input/output tokens per model call.

| METHOD | HWM | ACT. | PATCH | SUCC. | TOK/CALL |
|---|---|---|---|---|---|
| OURS | ✓ | ✓ | ✓ | **0.910** | 723 / **32** |
| −ACT. | ✓ | – | ✓ | 0.821 | 785 / 30 |
| −PATCH | ✓ | ✓ | – | 0.766 | **686** / 139 |
| HWM ONLY | ✓ | – | – | 0.697 | 759 / 140 |
| MEMGPT | – | – | – | 0.317 | 950 / 44 |

## 4.5. Step and Token Efficiency Analysis

We complement success rates with lightweight efficiency analyses derived from completed runs. All token counts include every model call in the agent loop (e.g., updater and executor calls when applicable, as well as summarization calls in summarization-based baselines).

### 4.5.1. AVERAGE STEPS TO TERMINATION

We report the average number of environment steps per episode, computed over all tasks. This statistic reflects how quickly a method tends to finish an episode, either by completing the task or by hitting an evaluation stopping criterion. We cap each episode by a maximum of $T = 50$ environment steps (step cap). In addition, we apply lightweight runtime guardrails uniformly across methods to prevent degenerate loops (e.g., excessive memory-edit iterations without environment progress). Such early stopping is rare in our runs; almost all failed episodes terminate at the step cap.

*Table 5.* Episode length statistics on ALFWorld `valid_unseen` (step cap $T = 50$). We report success rate (Succ.) and the mean-/median number of environment steps per episode over all tasks.

| METHOD | SUCC. | MEAN STEPS | MEDIAN STEPS |
|---|---|---|---|
| **QWEN3-4B (FP16)** | | | |
| REACT | 0.062 | 47.43 | 50 |
| SIMPLY SUMMARIZE | 0.228 | 44.01 | 50 |
| HIAGENT | 0.248 | 42.98 | 50 |
| MEMGPT | 0.317 | 33.90 | 50 |
| OURS | **0.910** | **19.83** | **17** |
| **GEMINI-2.5 FLASH** | | | |
| REACT | 0.786 | 27.28 | 25 |
| SIMPLY SUMMARIZE | 0.531 | 33.83 | 45 |
| HIAGENT | 0.559 | 29.63 | 26 |
| MEMGPT | 0.821 | 27.35 | 24 |
| OURS | **0.945** | **18.22** | **16** |

*Table 6.* Online token cost on ALFWorld `valid_unseen`. Tok/ep and Tok/act denote expected tokens per episode and per environment step, respectively; Tok/succ = Tok/ep divided by success rate (k = thousand tokens).

| METHOD | SUCC. | TOK/EP | TOK/ACT | TOK/SUCC |
|---|---|---|---|---|
| **QWEN3-4B (FP16)** | | | | |
| REACT | 0.062 | 328.6K | 6671 | 5294.1K |
| SIMPLY SUMMARIZE | 0.228 | 106.8K | 2336 | 469.3K |
| HIAGENT | 0.248 | 125.2K | 2750 | 504.2K |
| MEMGPT | 0.317 | 84.7K | 3079 | 267.1K |
| OURS | **0.910** | **36.2K** | **1848** | **39.8K** |
| **GEMINI-2.5 FLASH** | | | | |
| REACT | 0.786 | 146.3K | 4088 | 186.1K |
| SIMPLY SUMMARIZE | 0.531 | 58.1K | 1591 | 109.5K |
| HIAGENT | 0.559 | 35.3K | **1190** | 63.2K |
| MEMGPT | 0.821 | 39.0K | 1381 | 47.5K |
| OURS | **0.945** | **34.1K** | 1891 | **36.1K** |

### 4.5.2. TOKEN COST PER ACTION AND PER EPISODE

We report two deployment-oriented token metrics: expected tokens per episode (Tok/ep, including both successes and failures under the 50-step cap) and expected tokens per environment step (Tok/act). Tok/ep captures the end-to-end cost of attempting a task once, and can be dominated by methods with low success rates because many episodes run to the step cap. Tok/act partially normalizes for episode length and reflects per-decision context pressure. On Gemini, our method introduces a small constant per-step overhead due to structured updates and subgoal-conditioned interfaces, but achieves substantially higher task completion, making overall efficiency sensitive to success rates. We approximate token usage offline with `tiktoken` to obtain a consistent proxy across methods. This may differ from each model's native tokenizer/API accounting, but is sufficient for relative comparisons.

### 4.5.3. PER-STEP TOKEN GROWTH TREND

To visualize context pressure in long-horizon interaction, we plot the average tokens required for a single step decision as a function of step index $t \in [1, 50]$. At each step $t$, the average is computed over episodes that are still running at $t$ (episodes that already terminated are excluded). This highlights how different memory mechanisms control the growth of decision context over time.

Figure 3 illustrates per-step prompt token growth as interaction proceeds. **ReAct** shows near-linear prompt inflation as it accumulates raw history, quickly approaching the context budget, while **Simply Summarize** still exhibit gradual growth as summaries and retained context expand. **MemGPT** displays a characteristic periodic pattern, as its memory management mechanism triggers summarization

every five steps, resulting in a wave-like prompt growth curve. HiAgent also shows an increasing trend, since subgoals and their completion states must be retained in the interaction history. In contrast, **Ours** keeps the decision context bounded and relatively stable across steps, indicating better control of context pressure under long-horizon interaction. (Per-step averages are computed over episodes that are still running at step $t$.)

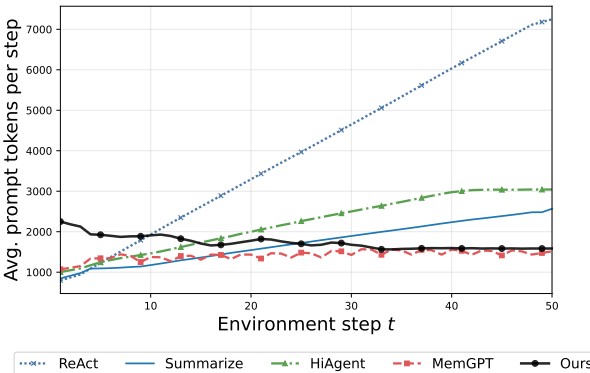

*Figure 3.* Per-step prompt token usage on ALFWorld `valid_unseen` (Qwen3-4B FP16). At each step $t$, we average over episodes that are still running at $t$.

### 4.5.4. TOKEN COST OF OFFLINE SCHEMA INDUCTION

Our method relies on an offline schema induction pipeline that builds a reusable, family-specific working-memory schema without any parameter updates. This schema is constructed once per task family and remains static during online execution. Starting from a manually designed base schema, we *incrementally* update the schema when incorporating an additional task family; the updated schema is then reused for all subsequent families and for all online episodes within each family. Specifically, the base schema is manually designed based on the simplest `pick-and-place` task in ALFWorld, and serves as the initial scaffold for subsequent schema expansion. For each new task family, we randomly sample five successful execution traces from the `valid-seen` split to perform offline schema induction. These traces are used exclusively in the offline stage and are not involved in online evaluation. We implement the LLM-based induction pipeline using GPT-5.1 as the induction LLM and report its token usage in Table 7. The one-time LLM induction cost ranges from $6.3 \times 10^4$ to $1.06 \times 10^5$ tokens per additional family, dominated by prompt (input) tokens, and is amortized across all instances of that family during online execution.

### 4.6. Failure Mode Analysis

We perform lightweight log-based analyses to better understand common failure patterns in long-horizon interaction.

*Table 7.* Incremental offline schema induction cost using GPT-5.1. We start from a base schema and sequentially update it when processing each additional task family; the resulting schema is reused thereafter. Tokens are reported as prompt (input), completion (output), and total.

| TASK FAMILY | INPUT TOKS | OUTPUT TOKS | TOTAL TOKS |
|---|---|---|---|
| LOOK−AT−LIGHT | 57,945 | 4,936 | 62,881 |
| CLEAN−PLACE | 69,217 | 4,182 | 73,399 |
| TWO−PLACE | 81,051 | 5,043 | 86,094 |
| HEAT−PLACE | 93,851 | 3,890 | 97,741 |
| COOL−PLACE | 101,887 | 4,122 | 106,009 |

First, we measure the *invalid action rate*, i.e., actions rejected by the environment due to grammar/parse failures or execution rejection. Table 8 shows that our structured, subgoal-conditioned action interface reduces invalid actions on both small and large backbones, suggesting improved grounding and action feasibility control.

*Table 8.* Invalid action rate (lower is better). We count actions rejected by the environment (parse failure or execution rejection).

| METHOD | QWEN3-4B (FP16) | GEMINI-2.5 FLASH |
|---|---|---|
| REACT | 30.0 | 16.5 |
| SUMMARIZE | 27.8 | 42.4 |
| HIAGENT | 33.3 | 18.2 |
| MEMGPT | 10.8 | 19.9 |
| OURS | **7.7** | **5.0** |

Second, we examine how unsuccessful episodes terminate under the evaluation protocol. Most failures across methods simply exhaust the 50-step budget without reaching a terminal success state, indicating that long-horizon completion (rather than early crashes) is the dominant challenge under weaker backbones. Beyond budget exhaustion, we observe two rare early-termination patterns. (i) On small backbones, MemGPT can enter a degenerate memory-edit loop, repeatedly issuing memory edits without environment progress until reaching step cap (26/145 episodes on Qwen3-4B FP16 in our runs). (ii) Our method exhibits a single premature termination due to an incorrect subtask completion update (1/145 episodes on Qwen3-4B FP16). These cases are infrequent overall and do not change the dominant failure profile, but they help contextualize occasional early exits observed in step statistics.

## 5. Related Work

**LM agents for long-horizon interactive tasks.** A common paradigm treats a language model as an interactive policy that repeatedly observes and acts, often interleaving intermediate reasoning with environment actions (e.g., ReAct; (Yao et al., 2023b)). Beyond direct prompting, many methods improve long-horizon performance by introduc-

ing explicit planning/decomposition or separating planning from execution (Wang et al., 2023a; Xu et al., 2023), and by searching over reasoning traces such as Tree/Graph-of-Thought style procedures (Yao et al., 2023a; Besta et al., 2024). Recent hierarchical and feedback-adaptive planning methods further decompose long-horizon problems into structured intermediate plans. HyperTree Planning constructs hypertree-structured planning outlines that iteratively refine complex problems into sub-problems (Gui et al., 2025), while Plan-and-Act separates high-level plan generation from low-level environment-specific execution (Erdogan et al., 2025). Adaptive planners such as AdaPlanner and DEPS revise plans based on environmental feedback, execution progress, or goal achievability in embodied and interactive environments (Sun et al., 2023; Wang et al., 2023b). Another influential line augments agents with external tools/APIs, expanding the effective action space via tool signatures and routing (Schick et al., 2023; Karpas et al., 2022; Patil et al., 2024). These approaches primarily target *how* the agent reasons, plans, or selects actions/tools. Our focus is complementary: under tight context budgets, we study *what* decision-relevant state must be retained across steps, maintaining a goal-conditioned decision-relevant information state and an approximation of the effective action set rather than relying on an ever-growing history.

**Agent memory and working-memory optimization.** To mitigate context growth, many systems compress interaction histories via free-form summaries, reflective rewrites, or iterative self-improvement (Madaan et al., 2023; Shinn et al., 2023). However, unconstrained rewriting may omit key facts or introduce distortions that compound over time. More structured memory management includes hierarchical subgoal-level compression (Hu et al., 2025a) and OS-style mechanisms that partition writable context and external buffers (Packer et al., 2024). Retrieval-augmented generation stores information externally and retrieves relevant snippets on demand (Lewis et al., 2020). Recent work also treats *working memory* as an optimization problem: compressing long contexts at inference time (Jiang et al., 2023; 2024), learning compact latent carriers or summarization tokens via training (Mu et al., 2023; Ge et al., 2024; Liao et al., 2025), and learning multi-turn consolidation policies via SFT/RL for long-horizon research or control (Yu et al., 2026; Zhou et al., 2026; Wu et al., 2025; Chen et al., 2025; Sun et al., 2025; Ye et al., 2026; Lumer et al., 2025). In contrast to both free-form summarization and learned consolidation, our method maintains a goal-conditioned, structured local workspace and updates it with rule-constrained patches, aiming to reduce state drift under small context windows while remaining training-free.

**Structured state carriers and schema-governed interfaces.** A growing trend in agent design is to externalize in-termediate computation into *structured artifacts*—schemas, checklists, program-like states, or subtask interfaces—to improve controllability and validation, rather than relying on opaque natural-language scratchpads. Related lines compile language into constrained policies or action primitives (Ichter et al., 2023; Liang et al., 2023), and induce reusable abstractions/programs from a small number of successful traces (Erdogan et al., 2024; Wölflein et al., 2025). Aligning with this direction, we define subtask interfaces with typed fields and guarded update rules, and instantiate them as local working memory during online execution. Online, state evolves via rule-constrained patch operations and action generation is conditioned on subtask-specific templates. This positions our approach as a training-free structured interface for stabilizing long-horizon execution, complementing both planning-centric agents and learning-based memory optimization.

## 6. Discussion

Our main takeaway is that SLM agents struggle in long-horizon interaction primarily due to poor contextual information efficiency. Rather than maintaining an ever-growing history, our framework makes decision-relevant state explicit by tracking (i) the active goal/subgoal, (ii) a subgoal-conditioned information state for progress assessment, and (iii) an approximate effective action set. This motivates a training-free hierarchical working memory: global memory carries task-level facts across subtasks, while a structured local interface directly conditions action selection and supports bounded, controllable updates.

The current framework also has obvious limitations. Our evaluation is concentrated on ALFWorld, a structured benchmark with reusable task-family procedures and observable subtask-completion signals. This setting matches the procedure-centric deployment scenario targeted by our method, but it does not by itself establish generalization to more open-ended environments with less stable task structure, branching, or rollback. In addition, our current implementation uses a fixed linear subtask workflow and does not yet include explicit recovery mechanisms when the agent misses critical evidence or incorrectly updates memory. Future work should extend the same schema-governed local-memory interface to richer control-flow structures and broader cross-benchmark evaluation.

## Impact Statement

This work aims to improve the deployability of agent systems by enabling small language models to perform long-horizon tasks with more information-dense and decision-relevant context, without costly post-training or online large-model calls. Potential benefits include lower inference cost

and latency, reduced dependence on centralized large-model serving, and more scalable deployment in on-device or high-concurrency settings. At the same time, maintaining explicit working memory introduces risks if the stored state contains sensitive information, is corrupted by adversarial observations, or becomes outdated and leads to incorrect automation. These risks are especially important when agents are connected to external tools or high-stakes workflows. Practical mitigations include minimizing and sanitizing stored content, enforcing access control and deletion policies, auditing memory updates, validating actions before execution, and retaining human oversight for consequential decisions.

## Acknowledgements

This work is supported by the National Natural Science Foundation of China under Grant Number 62402050.

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

# A. Implementation Details: Schema, Prompts, and Online Control Loop

## A.1. Field Mapping and Terminology

Our method maintains task-level *Global Memory* $M_G$ and subtask-level *Local Memory* $M_L$ (Table 2, Alg. 1). Within the appendix, we use the paper field names exclusively (CORE/CONTEXT/RETURN); implementation-specific key names are omitted for clarity.

| Field | Role in the paper (reviewer-facing) |
| --- | --- |
| **CORE** | Read-only subtask identity, goal, and natural-language termination condition (for the SLM). |
| **CONTEXT** | Online-writable minimal goal-conditioned information state used for decision making. |
| **RETURN** | Online-writable structured subtask outputs committed back to $M^G$ upon completion. |

*Table 9.* Core local-memory fields used throughout the appendix. ACTIONS and RULES are part of $\Sigma_F$ (read-only schema) and are injected into prompts.

## A.2. Task-Family Schema and Macro Initialization

A task family $F$ provides a schema $\Sigma_F$ that specifies: (i) a fixed subtask sequence $(g_1, \ldots, g_K)$ serving as an execution skeleton, and (ii) a macro task-state schema used for global memory $M_G$. The `MacroStateInitializer` runs once per episode and produces a JSON macro state. Initialization includes deterministic defaults (e.g., filling empty lists and inferring missing IDs from extracted receptacles) to remain robust when the initial observation is incomplete.

**Task families used in ALFWorld.** We evaluate on six families: *Pick-and-place-simple*, *Look-at-light*, *Clean-and-place*, *Heat-and-place*, *Cool-and-place*, and *Two-object-place*.

## A.3. Local Interface: Subtask Interfaces (Inputs, Actions, Rules, Termination)

Each subtask type $g_k$ is associated with a subtask interface $I_{g_k}$ containing:

- **Input parameters (static)** copied from macro state at subtask start via a control-layer caller.

- **Local interface schema** with `CORE`, `CONTEXT`, and `RETURN`.

- **Update constraints** that define (a) a typed patch grammar (operators), (b) an allow-list of writable JSON paths, and (c) additional guarded conditions.

- **Action templates and executor constraints** (read-only) used to condition action generation.

- **Termination condition** "done-when-all": the subtask ends once all required RETURN fields become non-empty.

## A.4. Rule-Constrained Patch Updates

At each environment step, the updater emits a list of patch operations that can modify **CONTEXT** and **RETURN** only; all other fields are read-only. Patches that violate the allow-list (paths/operators) are rejected. In our current implementation, representative operators include:

SET(path, value), LIST_REMOVE(path, value), LIST_APPEND_UNIQUE(path, value).

We additionally use lightweight guardrails to prevent unsupported updates (e.g., fabricating locations without explicit evidence in the latest observation).

## A.5. Prompts: Updater and Executor

We separate online execution into two single-turn SLM calls per environment step.

**State Updater.** The updater receives (CORE, CONTEXT, RETURN), the last action, and the latest observation, and outputs JSON-only patch operations that can modify CONTEXT and RETURN.

**Executor.** The executor reads the updated CONTEXT together with the schema-provided ACTION templates and emits one environment action, which is then checked against the environment grammar.

## A.6. Online Execution Loop (Implementation-Aligned)

---

**Algorithm 2** Online execution with schema-governed hierarchical memory (implementation-aligned).

---

1: **Input:** Goal $G$; family schema $\Sigma_F$ with subtask sequence $(g_1, \ldots, g_K)$; step cap $T$.
2: Receive initial observation $o_1$; set $t \leftarrow 1$.
3: $M_G \leftarrow \text{INITGLOBAL}(G, o_1; \Sigma_F)$ (MacroStateInitializer)
4: **for** $k = 1$ to $K$ **do**
5:    $M_L \leftarrow \text{INITLOCAL}(M_G, g_k; \Sigma_F)$ (Control-layer caller injects static inputs)
6:    **while** $\neg\text{DONE}(M_L.\text{RETURN})$ **and** $t \leq T$ **do**
7:       $\Delta \leftarrow \pi_{\text{upd}}(o_t, M_L)$ (StateUpdater: emits `patch_ops`)
8:       $M_L \leftarrow \text{APPLYPATCH}(M_L, \Delta, M_L.\text{RULES})$
9:       **if** $\text{DONE}(M_L.\text{RETURN})$ **then**
10:          **break**
11:       **end if**
12:       $a_t \leftarrow \pi_{\text{exe}}(M_L.\text{CONTEXT}, M_L.\text{ACTIONS})$ (Executor)
13:       Execute $a_t$; receive $o_{t+1}$; set $t \leftarrow t + 1$.
14:    **end while**
15:    $M_G \leftarrow \text{COMMIT}(M_G, M_L.\text{RETURN})$ (Control-layer return applier)
16: **end for**
17: **Output:** success/failure from the environment.

---

## A.7. Reproducibility Notes

We cap each episode by a maximum of $T = 50$ environment steps. Unless stated otherwise, we use greedy decoding (temperature 0, $top\_p = 1$, $n = 1$) together with the environment action grammar. We also apply lightweight runtime guardrails to prevent degenerate loops; such early stopping is rare compared to step-cap termination.

# B. Offline Schema Induction Example

This appendix gives a compact example of one offline schema-induction run for the ALFWorld task family `look_at_obj_in_light`. The purpose is to illustrate how trajectories are converted into a reusable family-level working-memory schema. We present the key information passed into each induction round and the outputs in excerpted form.

The induction example uses five successful teacher traces from the `valid-seen` split. The task label is treated as an opaque family identifier: successful terminal trace evidence has priority over the surface wording of the goal. In this family, the induced execution logic is:

$$\text{SEARCH\_OBJ\_AND\_LIGHT} \rightarrow \text{TAKE\_OBJECT} \rightarrow \text{USE\_LAMP}.$$

**Reading the example.** The trace excerpts below contain concrete ALFWorld runtime identifiers, such as `book 1` or `desklamp 1`. These identifiers illustrate how the induction abstracts instance-specific observations into reusable role-level fields, including `target_object`, `lamp_object`, `target_object_location`, and `lamp_object_location`. The prompt and output excerpts are lightly normalized for readability and controller-facing formatting, without changing the induced subtask structure, completion conditions, or update logic.

**Fixed induction interface.** In addition to successful trajectories, the induction prompt provides a fixed controller interface: allowed schema sections, field-naming conventions, patch operators, and the format of existing subtask interfaces. These constraints determine the output form, but not the family-specific subtask sequence, required state fields, completion predicates, or action patterns, which are inferred from trajectory evidence.

The existing subtask library available before this induction run contains: SEARCH_OBJECT, which searches receptacles for one target object location; TAKE_OBJECT, which takes a located object and updates inventory; and MOVE_OBJECT_TO_RECEP, which moves an inventory object to a target receptacle.

**Implementation terminology in prompt excerpts.** The prompt excerpts preserve the implementation-facing names used in our induction scripts. In these prompts, `task spec` denotes the task-level schema fragment within the family schema $\Sigma_{\mathcal{F}}$, including task-state fields, base subtask sequence, initialization rules, and caller mappings. Similarly, `subtask spec` denotes what the main text calls a subtask interface $I_g$, i.e., the controller-facing template used to instantiate local memory for a subtask. These implementation labels do not introduce additional abstractions.

### B.1. Stage 1: Trace-Grounded Task Model

Stage 1 asks the induction LLM to infer only what is supported by successful teacher traces and terminal success evidence.

---

**Input: Successful Trace Evidence and Environment Priors**

```
[SYSTEM]
You are analyzing successful ALFWorld teacher traces. Use the task label only as
an opaque identifier. Terminal successful trace evidence has priority over the
surface wording of the goal.

General environment priors:
- Objects and receptacles use exact runtime ids.
- To interact with an object, the agent must be adjacent to its containing
  receptacle.
- Some objects are immovable and must be used in place.
- Closed receptacles must be opened before their contents can be inspected.

[USER]
Task type: look_at_obj_in_light

Successful trace excerpts:

Trace A
goal: examine the alarmclock with the desklamp.
...
t=1 action: go to desk 1
observation: You arrive at desk 1. On the desk 1, you see ... a desklamp 1 ...
t=2 action: take desklamp 1 from desk 1
observation: Nothing happens.
...
t=6 action: take alarmclock 1 from desk 2
observation: You pick up the alarmclock 1 from the desk 2.
...
t=8 action: use desklamp 1
observation: You turn on the desklamp 1.

Trace B
goal: examine the book with the desklamp.
...
t=21 action: take book 1 from bed 1
observation: You pick up the book 1 from the bed 1.
t=24 action: use desklamp 1
observation: You turn on the desklamp 1.
```

---

**Output: Trace-Grounded Task Model**

```
{
  "success_conditions": [
    "The target object is in inventory.",
    "The agent is at the receptacle containing the lamp object.",
    "The lamp is turned on after a use action."
  ],
  "minimal_action_chain": [
    "Locate both the target object and the lamp.",
    "Take the target object into inventory.",
```

```
      "Navigate to the receptacle containing the lamp.",
      "Use the lamp."
    ],
    "required_state_fields": [
      "target_object",
      "lamp_object",
      "all_receptacles",
      "agent_position",
      "inventory",
      "target_object_location",
      "lamp_object_location",
      "lamp_status"
    ],
    "negative_evidence": [
      "Successful traces terminate on use desklamp 1, not on an examine action.",
      "Taking the lamp is ineffective and should not be part of the successful chain."
    ]
  }
```

This stage identifies the key task-specific fact that the final successful action is `use <lamp>` while the target object is already in inventory. It also identifies that the lamp should be treated as an immovable object to be used in place, rather than as an object to be picked up.

## B.2. Stage 2, Round 1: Library Coverage Check

The first synthesis round compares the trace-grounded task model against the existing subtask library and determines whether the task can be solved by existing subtasks.

**Input: Task Model and Existing Subtask Library**

```
[USER]
Given the trace-grounded model and the current subtask library, decide whether
the task can be solved by existing subtasks or requires minimal new capabilities.

Trace-grounded model:
- Required state: target_object, lamp_object, all_receptacles, inventory,
  target_object_location, lamp_object_location, lamp_status.
- Minimal chain: locate target and lamp, take target, navigate to lamp, use lamp.

Existing subtask library:
- SEARCH_OBJECT -> produces target_object_location for one target object.
- TAKE_OBJECT -> produces inventory.
- MOVE_OBJECT_TO_RECEP -> moves an inventory object to a destination receptacle.
```

**Output: Library Coverage and Missing Capabilities**

```
{
  "supported": "partial",
  "field_producers": {
    "target_object": "init",
    "lamp_object": "init",
    "target_object_location": "new",
    "lamp_object_location": "new",
    "inventory": "existing TAKE_OBJECT",
    "lamp_status": "new"
  },
  "missing_capabilities": [
    "A combined search capability must discover both the target object location
     and the lamp location.",
    "A use-lamp capability must issue use <lamp_object_id> at the lamp
     receptacle and record lamp_status."
```

```
    ],
    "minimal_delta": [
      "Add SEARCH_OBJ_AND_LIGHT.",
      "Add USE_LAMP."
    ]
 }
```

This round avoids a failure mode in which a single-object search locates only the target object and never records the lamp location. The output therefore introduces a minimal new search subtask that jointly searches for both required entities.

### B.3. Stage 2, Round 2: New Subtask Interface Synthesis

The second synthesis round expands the missing capabilities into executable subtask specifications.

---

**Input: Missing Capabilities and Subtask-Spec Format**

```
[USER]
Create minimal new subtask specs for the missing capabilities. Follow the
existing subtask format:
- caller_mapping maps task_state fields into subtask memory.
- operation_space emits JSON patch_ops.
- done_when_all lists return fields required for completion.

Coverage report:
- Need SEARCH_OBJ_AND_LIGHT to return target_object_location and
  lamp_object_location.
- Need USE_LAMP to return lamp_status.
```

---

**Output: New Subtask Interfaces**

```
{
  "new_subtasks": [
    {
      "subtask_type": "SEARCH_OBJ_AND_LIGHT",
      "caller_mapping": {
        "target_object": "target_object",
        "lamp_object": "lamp_object",
        "all_receptacles": "unsearched_receptacle_ids"
      },
      "return": {
        "target_object_location": null,
        "lamp_object_location": null
      },
      "allowed_patch_ops": [
        "SET(return.target_object_location, <current_receptacle_id>)
         if the observation contains target_object at the current receptacle.",
        "SET(return.lamp_object_location, <current_receptacle_id>)
         if the observation contains lamp_object at the current receptacle.",
        "LIST_REMOVE(context.unsearched_receptacle_ids,
         <current_receptacle_id>) after the receptacle has been inspected."
      ],
      "base_actions": [
        "go to <receptacle>",
        "open <receptacle>"
      ],
      "done_when_all": [
        "return.target_object_location",
        "return.lamp_object_location"
      ]
    },
    {
      "subtask_type": "USE_LAMP",
```

```
        "caller_mapping": {
          "lamp_object": "lamp_object",
          "lamp_object_location": "lamp_object_location",
          "inventory": "inventory"
        },
        "return": {
          "lamp_status": null
        },
        "allowed_patch_ops": [
          "SET(return.lamp_status, on)
           if the observation confirms that the lamp has been turned on."
        ],
        "base_actions": [
          "go to <receptacle>",
          "use <lamp_object_id>"
        ],
        "done_when_all": [
          "return.lamp_status"
        ]
      }
    ]
  }
```

The patch operations define the writable surface for online execution. During evaluation, patches outside the allowed operation space or patches that attempt to modify read-only fields are rejected by the controller.

### B.4. Stage 2, Round 3: Task State and Base Sequence

The third synthesis round assembles the task-level state schema and base subtask sequence. The prompt explicitly requires search-discovered locations to be produced by earlier subtasks rather than initialized directly.

---

**Input: Candidate Subtasks and Construction Constraints**

```
[USER]
Construct the task spec for look_at_obj_in_light.

Use:
- trace_grounded_model.minimal_action_chain
- coverage_report
- new_subtasks from Round 2
- existing mature task spec examples only for schema/style consistency

Constraints:
- Search-discovered locations must be produced by subtasks, not initialized.
- The base sequence must only use fields that are initialized or produced by
  earlier steps.
- Do not add an examine terminal step unless supported by terminal success traces.
```

---

**Output: Task State Schema and Base Sequence**

```
{
  "task_type": "look_at_obj_in_light",
  "task_state_schema": {
    "fields": {
      "goal": "string",
      "target_object": "string",
      "lamp_object": "string",
      "all_receptacles": "list[string]",
      "agent_position": "string|null",
      "inventory": "list[string]",
      "target_object_location": "string|null",
```

```
        "lamp_object_location": "string|null",
        "lamp_status": "string|null"
      }
    },
    "base_subtask_sequence": [
      {
        "name": "searchBoth",
        "subtask_type": "SEARCH_OBJ_AND_LIGHT",
        "expects": ["target_object", "lamp_object", "all_receptacles"],
        "produces": ["target_object_location", "lamp_object_location"]
      },
      {
        "name": "takeTarget",
        "subtask_type": "TAKE_OBJECT",
        "expects": ["target_object", "target_object_location"],
        "produces": ["inventory"]
      },
      {
        "name": "useLamp",
        "subtask_type": "USE_LAMP",
        "expects": ["lamp_object", "lamp_object_location", "inventory"],
        "produces": ["lamp_status"]
      }
    ]
}
```

This is the key round that fixes the high-level execution logic. The first subtask searches for both the target object and the lamp, ensuring that the subsequent USE_LAMP subtask receives a valid lamp location.

### B.5. Stage 2, Round 4: Caller Mapping and Initialization

The fourth synthesis round verifies how task-level fields are passed into subtasks and proposes initialization rules.

---

**Input: Task Fields and Base Sequence**

```
[USER]
Given the task spec and subtask specs, verify how task_state fields are passed
into each subtask. Also propose task init_rules.

Task fields:
goal, target_object, lamp_object, all_receptacles, agent_position, inventory,
target_object_location, lamp_object_location, lamp_status

Base sequence:
SEARCH_OBJ_AND_LIGHT -> TAKE_OBJECT -> USE_LAMP
```

---

**Output: Caller Mappings and Initialization Rules**

```
SEARCH_OBJ_AND_LIGHT:
  target_object -> target_object
  lamp_object -> lamp_object
  all_receptacles -> unsearched_receptacle_ids

TAKE_OBJECT:
  target_object -> target_object
  target_object_location -> target_object_location
  inventory -> inventory_snapshot

USE_LAMP:
  lamp_object -> lamp_object
  lamp_object_location -> lamp_object_location
  inventory -> inventory
```

---

The important induced behavior does not depend on initializing search-discovered locations. Both `target_object_location` and `lamp_object_location` remain dynamic fields produced by `SEARCH_OBJ_AND_LIGHT`.

## B.6. Stage 2, Round 5: Final Patch Check

The final round checks the candidate task schema, new subtask interfaces, and caller mappings before materializing the controller-facing schema fragments.

---

**Input: Candidate Schema for Integration Check**

```
[USER]
Review the candidate task spec, new subtask specs, and Round 4 mappings. Return
any final task_spec_patch or subtask_spec_patches needed for controller integration.
```

---

**Output: Final Controller-Facing Patch**

```
{
  "subtask_spec_patches": {},
  "task_spec_patch": {
    "init_rules": [
      "goal := <episode_goal>.",
      "target_object := extracted_target_object_from_goal.",
      "lamp_object := \"desklamp\".",
      "all_receptacles := <episode_receptacle_ids>.",
      "agent_position := null.",
      "inventory := [].",
      "target_object_location := null.",
      "lamp_object_location := null.",
      "lamp_status := null."
    ]
  }
}
```

---

Here, `<episode_receptacle_ids>` denotes instance-level initialization from the current environment observation or metadata. The schema stores the field `all_receptacles`; it does not hard-code a particular set of receptacle ids across episodes. Likewise, target and lamp locations are not initialized and must be filled by the search subtask during online execution.

## B.7. Integrated Schema Fragment

The accepted round-level outputs are then combined into controller-facing schema fragments. The integrated task fragment contains the following task-level fields:

```
target_object
lamp_object
all_receptacles
agent_position
inventory
target_object_location
lamp_object_location
lamp_status
```

and the following base subtask sequence:

```
SEARCH_OBJ_AND_LIGHT -> TAKE_OBJECT -> USE_LAMP
```

The integrated files simply materialize the accepted round-level specifications into the final schema format used by the controller. The round-level excerpts above show the trace evidence and synthesis decisions leading to these fragments.

Overall, this example illustrates that, for the `look_at_obj_in_light` family, five successful teacher traces together with general environment priors provide enough evidence for the induction pipeline to recover the task-specific decomposition and decision-relevant state fields used by our implementation.

