# OpenReview forum: "LightWM: Training-Free Hierarchical Working Memory for Small Language Model Agents"
_ICML.cc/2026/Conference — ICML 2026 regular_

### Official Review · Reviewer_N6FM · 2026-03-12

**Soundness:** 2
**Presentation:** 3
**Significance:** 3
**Originality:** 2
**Overall Recommendation:** 3
**Confidence:** 3

**Summary:**

This paper focuses on a practical deployment problem of small language model agents - how to accomplish long-horizon tasks without finetuning. It introduces a complete solution based on multiple techniques including hierarchical memory + structured schema + rule-constrained patch updates + offline family-level schema induction, and presents convincing results on performance and efficiency improvements on the ALFWorld benchmark.

**Compliance With Llm Reviewing Policy:**

Affirmed.

**Final Justification:**

Thank the authors for the detailed response. I especially appreciate the added ablation. However, I still think more experiments beyond ALFWorld are needed to prove the generalization ability and applicability. I therefore keep my current score.

**Key Questions For Authors:**

1. Almost all experiments were conducted on onducted on the valid_unseen split of ALFWorld, and the models used are mainly Qwen3‑4B/8B (+ Gemini‑2.5 Flash). Could you please explain: Which task types do you believe this training‑free hierarchical working memory framework is effective for (e.g., embodied environments, multi‑step tool use, long‑document reasoning, etc.)? Beyond ALFWorld, have you performed any preliminary experiments on other environments or task families? If yes, please provide the results or at least a qualitative discussion. If no, please more clearly define the applicable boundaries of this work and explain why using only ALFWorld is sufficient to support the main conclusions.
2. An important claim in the paper is that the primary reason small models fail in long‑horizon tasks is not insufficient reasoning ability, but rather low contextual information efficiency. The evidence of this claim mainly comes from comparisons of success rate and efficiency under different memory frameworks. We ask the authors to further elaborate: Have you conducted additional ablation experiments—such as gradually simplifying or removing fields like CONTEXT, RULES, RETURN in the local memory, or relaxing patch rules—to validate the marginal contribution of each component to performance?
3. The approach in this paper relies on one‑shot offline schema induction for each task family, assuming access to a small number of high‑quality successful trajectories. It would be good if the authors provide further discussion on the following points: When successful trajectories themselves contain noise or reflect suboptimal policies, will the induced schema perpetuate these flaws? Have you observed the phenomenon where a bad schema causes systematic failures of the entire system? In scenarios where the task distribution is more complex and task family boundaries are ambiguous—such as real‑world business systems with numerous task combinations and variants—does this per‑family schema design remain practical? Does it require some form of online schema adaptation, merging, or splitting mechanism to maintain scalability?

I am open to raising my scores if the above questions are adequately answered.

**Limitations:**

Yes

**Strengths And Weaknesses:**

**Strengths:**
- (Soundness) The method proposed in this paper has a clear theoretical motivation: From the POMDP perspective, it explicitly points out that the key information needed for decision making is the current subgoal, the state relevant to progress evaluation, and the current set of available actions. The hierarchical memory mechanism is designed based on this.
- (Significance) It is a long-standing challenge to support more powerful agents with smaller models.
- (Originality) I think the engineering efforts and architectural design of the SLM agent memory is original.

**Weaknesses:**
- (Soundness) The evaluation is limited to a single benchmark ALFWorld, and not tested in other benchmarks/environments (e.g., other embodied, computer use, etc.). Which does not sufficiently demonstrate generalization ability.
- (Originality) The methods introduced in this paper have demonstrated good engineering efforts, but the theoretical insights and understanding behind the methods are not clear.

---

> ### Author Rebuttal · Authors · 2026-03-31
>
> # Q1: Applicable Task Types and Evaluation Sufficiency
>
> Our framework targets procedural agent tasks with three structural properties: (P1) family-level workflow regularity enabling a reusable planning skeleton; (P2) observation-grounded subtask completion mappable to extractable information-state fields; (P3) parameterized action output rather than open-ended generation. This structural profile corresponds precisely to the high-frequency, execution-oriented subtasks identified as a core deployment value of SLMs [1].
>
> ALFWorld's internal diversity already covers the structural variations our framework targets. The six families span single-step relocation to multi-object long-horizon coordination, and only one schema (pick-place) was manually designed—the remaining five were independently induced by our offline pipeline, evidencing generalization of the induction mechanism rather than family-specific tuning.
>
> | Task Family   | Complexity             | N   | Ours  | MemGPT | ReAct |
> | ------------- | ---------------------- | --- | ----- | ------ | ----- |
> | Pick-place†   | Single relocation      | 24  | 87.5% | 50.0%  | 0.0%  |
> | Look-at-light | State switching        | 18  | 100%  | 50.0%  | 11.1% |
> | Clean-place   | Intermediate transform | 31  | 83.9% | 41.9%  | 16.1% |
> | Heat-place    | Intermediate transform | 27  | 92.6% | 29.6%  | 7.4%  |
> | Cool-place    | Intermediate transform | 27  | 96.3% | 14.8%  | 0.0%  |
> | Two-object    | Longest horizon        | 18  | 88.9% | 0.0%   | 0.0%  |
>
> †Base schema manually designed; remaining five induced by our offline pipeline.
>
> Beyond ALFWorld, the framework's planning structure is not restricted to fixed linear subtask sequences. Richer control structures are natural extensions of the same schema-governed interface, which we identify as a promising direction for broader deployment settings such as multi-step tool use and document-centric tasks.
>
> ---
>
> # Q2: "Contextual Information Efficiency" Claim and Ablation Evidence
>
> We provide new ablations (all 145 valid_unseen tasks, Qwen3-4B FP16) that directly validate our core claim and quantify the marginal contribution of each component. Our framework comprises three components: ① hierarchical goal-conditioned memory representation, ② action templates, and ③ patch constraints.
>
> The key finding is that component ① alone—without any action pruning or patch constraints—brings the same 4B model to 69.7% success, more than double MemGPT's 31.7%. Our results show that improving information organization alone enables a 4B model to recover the majority of the performance gap, without modifying its reasoning capability.
>
> |Variant|Success|Input Tok/call|Output Tok/call|
> |---|---|---|---|
> |Full (①+②+③)|91.0%|723|32|
> |−② action templates|82.1%|785|30|
> |−③ patch constraints (→ full rewrite)|76.6%|686|139|
> |−②−③ (representation only)|69.7%|759|140|
> |MemGPT (strongest baseline)|31.7%|950|44|
>
> ---
>
> # **Q3: Schema Robustness and Applicability Boundaries**
>
> **Stage design enables noise filtering.** Our offline induction pipeline first elicits shared completion conditions, information-state variables, and the minimal action set across multiple traces, then synthesizes these into a conformant schema and resolves structural inconsistencies. Idiosyncratic behaviors in individual traces lack cross-trace consistency and are naturally discarded during generalization.
>
> **The real risk is selection bias, and it produces detectable signals.** When all sampled traces happen to share an initial configuration that already satisfies part of the success conditions, the induction LLM cannot distinguish these pre-satisfied conditions from genuinely irrelevant ones, leading to an incomplete schema. For instance, if sampled look-at-light traces all start with the lamp co-located, the schema may incorrectly assume the agent must be holding the object rather than navigating to the lamp. Crucially, such errors manifest as family-wide systematic failures rather than sporadic instance-level mistakes, making them straightforward to detect and correct through a lightweight validation pass.
>
> **Complex task distributions call for richer planning structures, not schema merging.** A task family is defined by a shared planning structure; where task structures are genuinely distinct, the appropriate response is to design DAG- or FSM-style workflows rather than merge families under a single fixed sequence. For large systems with dynamic task combinations, our framework is well-suited as a structured SLM sub-component within a larger LLM-orchestrated architecture, handling high-frequency routine subtasks at bounded cost.
>
> ---
>
> We thank the reviewer for the constructive feedback. We will release code and all induced schemas upon publication. We welcome further discussion during the rebuttal period.
>
> ---
> # Reference
>
> [1] Belcak et al., "Small Language Models are the Future of Agentic AI", arXiv 2025

---

> > ### Author Rebuttal · Reviewer_N6FM · 2026-04-04
> >
> > Thank the authors for the detailed response. I especially appreciate the added ablation.
> > However, I still think more experiments beyond ALFWorld are needed to prove the generalization ability and applicability.

---

> > > ### Author Response · Authors · 2026-04-08
> > >
> > > We thank the reviewer for the direct feedback and for acknowledging the value of the ablation results. We also want to express genuine appreciation for the depth and constructiveness of the questions raised in the first round.
> > >
> > > We fully agree that broader empirical validation is important and are actively extending the framework in that direction. We have begun preliminary experiments on environments such as ScienceWorld, but cannot report complete results within the rebuttal window. On the design side, the explicitly structured fields in local memory naturally accommodate more complex control flow, though richer planning structures remain a key challenge for broader generalization—this is a primary focus of our ongoing work.

---

### Official Review · Reviewer_qQSU · 2026-03-12

**Soundness:** 3
**Presentation:** 3
**Significance:** 4
**Originality:** 3
**Overall Recommendation:** 5
**Confidence:** 4

**Summary:**

This paper develops a hierarchical working memory system for the purpose of enhancing small language model agent performance over long interactions. This training-free memory framework is divided into a task-level global memory and a subtask-level local memory, which leverages a structured memory schema defined offline by a  LLM over specific task families.  Online memory modifications are handled through rule-constrained patches issued by the SLM, while agent action selection is conditioned on subtask-level templates. The authors evaluate this framework on the ALFWorld valid_unseen dataset for text-aligned embodied reasoning, showing that their framework leads to substantial increases in task success rate over other summarization-based approaches when evaluated on three models: Qwen3-4B, Qwen3-8B, and Gemini-2.5-Flash. The efficiency of their framework is further demonstrated through analysis on token usage, steps to task completion, and failure-rate showing significant lower rates for their framework as compared to other memory-augmented baselines.

**Compliance With Llm Reviewing Policy:**

Affirmed.

**Final Justification:**

The approach is original and shows impressive improvements over existing small language models on similar tasks. The work could benefit from further experiments on a wider set of tasks to show generalizability, but I believe what is presented here is convincing enough to show the current work has merit.

**Key Questions For Authors:**

1. There is a significant lack of detail on the offline phase of schema induction. How do you prompt the external LLM to determine subtask sequences and requirements? This is a critical component of the framework that is not explored well.

2. It is mentioned that only a handful of successful task trajectories are used to define the structured task/subtask schema. Do you have any analysis on task accuracy dependence on the number of trajectories use to produce the schema? This would be a good addition to the supplementary material.

3. You do briefly mention your evaluation being limited to the ALFWorld embodied tasks. How readily applicable is your framework to other agent task domains? Could your structured task memory formulation also be used to improve performance of LLMs on repetitive knowledge tasks?

**Limitations:**

Yes (although limitations are mentioned very briefly).

**Strengths And Weaknesses:**

**Soundness/Presentation:**

The authors' methods do a nice job of detailing the two-tiered memory hierarchy in their framework design. The framework utilizes a global memory that contains structured task-level information, and a local memory which maintains a subtask scheme through which the SLM chooses actions and issues updates to global memory. The overall memory systems is updated through an offline phase, in which a secondary LLM is used to produce the task family scheme, and an online phase, in which the SLM uses rule-constrained patch updates to the global memory. In general, the online phase is clearly outlined and easily understood. However, the offline schema formulation is critical to the memory framework, and few details are given as to how this is accomplished. The local memory fields are detailed nicely, so we can infer how the offline phase may be accomplished, but it seems that more details and perhaps an example should be provided outlining this offline phase. Given the main task being performed, the evaluation metrics (accuracy, number of steps, token count, invalid action frequency, etc.) appear appropriate.

The experimental design is straight-forward, involving evaluating task performance on the ALFWorld valid_unseen split for six tasks with Qwen3-4b, Qwen3-8B, and Gemini-2.5 Flash. The evaluation is somewhat limited (only using a single split within a single dataset), but it is understandable that there are limited options for benchmarks given the authors' intentions (on-board deployment with SLMs) for the memory framework. However, this memory framework does not seem limited to the embodied reasoning task and could be extended to other agent task execution benchmarks like ScienceWorld or WebShop. Given that this is a training-free method, it would also be nice to include some models outside of the Qwen family as well.

**Significance:**

The exceptional significant increase in success rate on the ALFWorld benchmark could position this framework as a major contribution to decision-making on long contexts. However, the framework heavily relies on the task/subtask schema produced offline by a closed-source LLM, limiting the flexibility of this framework. Additional details on the offline phase should be provided, and (time-permitting) perhaps some additional experiments using an open-weight model for task schema production. Nonetheless, even within a limited application scope, this memory framework could be a significant advancement for on-board decision-making.

**Originality:**

This work is original in its introduction of a structured task schema to a hierarchical memory store, specifically for on-board deployment of SLMs for embodied reasoning.

---

> ### Author Rebuttal · Authors · 2026-03-31
>
> # Q1: Offline Schema Induction Details
> Our induction is a **multi-stage structured pipeline** that extracts task structure from a small set of successful trajectories and maps it into a reusable schema.
>
> In **Stage 1**, the LLM performs trajectory-grounded structure extraction, identifying:
> - a minimal subtask sequence (execution skeleton),
> - required information-state variables for progress tracking
> - state transitions grounded in observable evidence.
>
> For example, given trajectories for a “heat-then-place” task, Stage 1 may extract a subtask sequence such as "SEARCH_OBJECT → TAKE_OBJECT → HEAT_OBJECT → PLACE_OBJECT",
> together with state variables like {target_object_location, heated_status} and update rules (e.g., setting heated_status=true after observing a successful heat action).
>
> In **Stage 2**, these extracted structures are mapped into our working-memory schema via constrained synthesis, aligning them with fixed schema roles (CORE / CONTEXT / ACTIONS / RULES / RETURN) and enforcing a minimality principle.
>
> We are happy to provide further clarification and discussion if helpful, and will release all detailed artifacts upon publication.
> # Q2: Trajectory Count Sensitivity
>
> The role of multiple trajectories in our offline induction is to recover **shared structure and decision-relevant state abstractions** for a task family, rather than instance-specific behaviors.
>
> Concretely, using multiple traces helps (i) identify a consistent subtask sequence , (ii) extract stable information-state variables required for progress tracking, and (iii) reduce noise from individual trajectories. With too few traces, the induced schema may overfit to a single instance and fail to capture the common planning structure across the family.
>
> As the number of traces increases, these shared patterns become more reliably identified, leading to improved schema quality. However, diminishing returns are expected once the key subtask structure and state variables are sufficiently covered, after which performance is inherently bounded by the expressiveness of the schema representation.
>
> In our experiments, we use 5 randomly sampled successful trajectories per task family, which we find sufficient to consistently recover the shared structure and achieve strong performance on ALFWorld.
>
> ---
>
> # Q3: Applicability to Other Domains
>
> Our framework targets **procedure-oriented agent tasks** that admit reusable, family-level structure. In particular, it assumes that (1) tasks share a common planning structure, (2) subtask completion can be determined from observable signals, and (3) actions can be expressed as parameterized commands.
>
> ALFWorld satisfies these properties and serves as a controlled testbed. More broadly, properties (2) and (3) hold for many agent domains. Extending our framework to richer settings primarily requires generalizing the planning structure beyond fixed subtask sequences. For example, WebShop-style tasks may require loops and backtracking, while ScienceWorld involves compositional and branching procedures.
>
> For repetitive knowledge-based tasks (e.g., structured editing or compliance workflows), similar procedural patterns often exist. With appropriate definitions of completion signals and task structure, our schema-based formulation may also be applicable.
>
> ---
>
> We thank the reviewer for the insightful questions. We will release all code, induction prompts, and induced schemas upon publication.

---

> > ### Author Rebuttal · Reviewer_qQSU · 2026-04-02
> >
> > I thank the authors for their work and the thought put into their rebuttal. I appreciate the clarification on schema induction; it may be helpful to include a concrete example of this induction process on a ALFWorld task in the Appendix. As to my Q2 (and partly Q3 as well), it seems that ALFWorld tasks are sufficiently similar enough that the current schema induction method works well with relatively few trajectories, but I'm curious how robust this approach would be on more complex/diverse tasks. Regardless, I believe this work has clear merits and plenty of room for future extensions/improvements, and will update my score accordingly.

---

> > > ### Author Response · Authors · 2026-04-08
> > >
> > > We thank the reviewer for the thoughtful follow-up and for the encouraging assessment of our work. We are glad that our clarifications on schema induction were helpful.
> > >
> > > **Appendix example.** We will include a concrete, step-by-step schema induction example for an ALFWorld task family in the Appendix, as suggested.
> > >
> > > Regarding robustness on more complex tasks, we offer some further thoughts:
> > >
> > > **Online-phase extensibility.** Greater task complexity primarily manifests as more subtasks and richer control flow (e.g., branching, backtracking). While our current implementation uses a fixed linear sequence, the RETURN-based completion boundary naturally extends to richer control flow: transition conditions can be defined over RETURN field states to route between subtasks or trigger re-entry when required fields remain unfilled.
> > >
> > > **Implications for offline induction.** Accordingly, induction prompts can be extended to elicit macro-level control-flow structures (branching conditions, retry triggers) alongside the current subtask specifications. The micro-level schema design—CONTEXT/RETURN fields, update rules, and action templates within each subtask—remains applicable, as these are governed by subtask-type semantics rather than inter-subtask ordering.
> > >
> > > **Broader evaluation.** As the reviewer rightly notes, ALFWorld's procedural regularity makes schema induction tractable with few trajectories. We are actively extending to more diverse benchmarks to stress-test this; complete cross-benchmark results are beyond the rebuttal window, but this is a key direction of our ongoing work.
> > >
> > > We appreciate the reviewer's recognition that this work has clear merits and room for future extensions, and we look forward to pursuing these directions.

---

### Official Review · Reviewer_PF6Q · 2026-03-13

**Soundness:** 3
**Presentation:** 3
**Significance:** 2
**Originality:** 2
**Overall Recommendation:** 3
**Confidence:** 4

**Summary:**

The paper introduces a training-free hierarchical working memory framework for Small Language Models (SLMs) to tackle long-horizon, interactive tasks. Instead of relying on expanding context windows or unstructured natural-language summarization, the method decomposes tasks into a global task-level memory and a subtask-level local memory. An offline LLM induces a goal-conditioned schema per task family, which defines a fixed subtask sequence, relevant variables, and valid action templates. Online, the SLM acts as a "State Updater" (generating rule-constrained memory patches) and an "Executor" (generating actions from local memory).

**Compliance With Llm Reviewing Policy:**

Affirmed.

**Key Questions For Authors:**

Please refer to the weaknesses

**Limitations:**

yes

**Strengths And Weaknesses:**

[Strengths]
Boosting a 4B parameter model from a 0.062 to a 0.910 success rate on ALFWorld is promising.

Transitioning from free-form natural language summarization (used in MemGPT or HiAgent) to explicit, rule-constrained patch operations (e.g., SET, LIST_REMOVE) seems to be feasible and reasonable.

The separation of offline LLM schema induction and online SLM execution is a good combination of existing capabilities.

The framing of the working memory as an approximate decision-relevant information state within a POMDP is technically reasonable.

[Weaknesses]
I am a little confused by the strict Markovian assumption in the State Updater. Because raw history is discarded entirely in favor of the current observation and local memory, it is not entirely clear how the agent recovers if the SLM fails to extract a crucial fact at step t or hallucinates a patch.

The authors seek to discuss the aspect of explicit correction and backtracking in Section 6, but could you clarify how the framework currently handles environment dynamics that force a regression (e.g., accidentally dropping an item, requiring a return to a "search" subtask)?

Could you provide more detail regarding the baseline comparisons, specifically concerning the "Pruned Grammar" module? The proposed framework restricts the SLM to a highly constrained, subtask-specific action space, but it appears baselines like ReAct are evaluated on the full environment grammar.

Since 4B models notoriously struggle to output perfectly formatted JSON-style commands (e.g., LIST_APPEND_UNIQUE) zero-shot, could you specify how syntax errors or invalid patches were handled online?

I wonder if the authors could elaborate on the expressivity of the offline schema for more complex domains. The assumption that tasks can be universally modeled via static execution skeletons seems somewhat restrictive for POMDPs requiring dynamic branching or conditional logic (e.g., "if door is locked, find key; else open").

---

> ### Author Rebuttal · Authors · 2026-03-31
>
> # Error Handling in Local Memory State Updates
>
> **Syntax-level failures are rare and empirically harmless in our setting.** Concretely, in our evaluation logs, malformed updates occur in only about 3% of state-update attempts, and all of them are benign no-op cases caused by misplaced quotation marks. Once rejected by the controller, the memory state remains unchanged, and we observe no measurable impact on final task success.
>
> **Semantic errors are controlled by a schema-governed writable surface.** Only `CONTEXT` and `RETURN` are writable online; all other fields are read-only, and any patch outside the allow-list is rejected.
>
> **We keep recovery mechanisms minimal to preserve clean attribution to the proposed framework.** The key value of our design is to constrain online state updates to decision-relevant fields in a lightweight and controllable manner. More elaborate recovery mechanisms, such as evidence extraction, rollback, or verifier-style checks, are promising future directions, but they are outside the scope of this work.
>
> # Subtask Sequence Expressivity
>
> **The fixed linear subtask sequence is motivated by highly procedural agent task families.**  These tasks involve reusable interaction procedures and limited variation across instances, making them a natural setting for efficient SLM-based agents. This perspective is also consistent with recent discussions on agentic workloads of Nvidia[1], which emphasize repeated execution of a small set of specialized procedures rather than fully general reasoning.
>
> **Our contribution lies in the schema-governed hierarchical working memory rather than the specific execution skeleton.**  The fixed subtask sequence serves only as an execution scaffold that defines subtask boundaries and information flow, while our main contribution is the structured, goal-conditioned working memory with hierarchical organization and rule-constrained updates. As discussed in Section 6, more expressive control flow such as branching or rollback is compatible with this design but beyond the current scope.
>
> **While the subtask sequence is fixed, the framework generalizes across diverse instance-level configurations.** ALFWorld valid unseen is specifically designed to test this: each instance differs in object positions, receptacle layouts, and interaction sequences. The consistent success across 145 unseen episodes demonstrates that our schema captures reusable decision structure without overfitting to specific instance configurations.
>
> # Action Space Pruning and Evaluation Fairness
>
> We appreciate the reviewer’s concern regarding fair attribution of gains and agree that this is an important factor to control for.
>
> **The main performance gain does not come from the restricted action space.**
> Replacing subtask-specific action templates with the full environment grammar reduces success from 91.0% to 82.1%, which remains over 2.6× higher than MemGPT (31.7%). Further removing patch constraints yields 76.6%, and removing both yields 69.7%, showing that the majority of gains come from the schema-governed hierarchical memory, while action templates provide a bounded +8.9pp improvement.
>
> ---
>
> More broadly, we believe the core contribution—making decision-relevant state explicit through a goal-conditioned, schema-governed local interface—is a conceptually general principle for constraining SLM execution in procedural agent settings. The fact that a 4B model under this design matches or exceeds LLM-level baselines in a competitive benchmark, without any parameter updates or online LLM calls, reflects a meaningful step toward practical, low-cost agent deployment.
>
> We are happy to clarify any of the above points further. Code and induced schemas will be released upon publication.
>
> [1] Belcak et al., "Small Language Models are the Future of Agentic AI", arXiv 2025

---

> > ### Author Rebuttal · Reviewer_PF6Q · 2026-04-06
> >
> > Thank you for the detailed rebuttal and clarifications. However, I still have lingering concerns regarding the lack of mechanisms for error recovery and backtracking, the limited expressivity of the static schemas for handling dynamic branching, and the reliance on heavy action space constraints compared to the baselines. I will keep my current rating.

---

> > > ### Author Response · Authors · 2026-04-08
> > >
> > > We sincerely thank the reviewer for the continued discussion. We appreciate the reviewer's insights on error recovery and dynamic branching, which provide valuable perspectives for future extensions. We will explicitly incorporate these discussions into our revised manuscript.
> > >
> > > To clearly conclude our stance on the remaining concerns, we would like to briefly summarize our position:
> > >
> > > **1. Error Recovery and Branching are Orthogonal to State Representation**. We agree that dynamic branching and backtracking are essential for handling complex environments. However, these are control-flow mechanisms that operate on top of state representations. Our core contribution in this work focuses on establishing the structured, goal-conditioned working memory itself. As noted in our initial response, the current linear sequence serves as an execution scaffold to demonstrate this memory framework. Integrating more expressive control flows remains a promising direction.
> > >
> > > **2. Unconstrained Evaluation Confirms a +50.4pp Gain**. We reiterate that our performance is driven by the memory framework, not action constraints. In our ablation study, when we exposed the **full environment grammar** to the SLM (completely removing the restricted action templates), our method still achieved an **82.1%** success rate. This represents a **+50.4pp** absolute improvement over the strongest baseline (MemGPT at 31.7%), demonstrating that our comparison is fair and the memory design is the primary performance driver.
> > >
> > > We thank the reviewer again for constructive feedback throughout the review process.

---

### Official Review · Reviewer_9N4f · 2026-03-13

**Soundness:** 3
**Presentation:** 3
**Significance:** 2
**Originality:** 3
**Overall Recommendation:** 4
**Confidence:** 4

**Summary:**

The paper proposes a training-free hierarchical working memory architecture for small language model agents that separates global task information from subtask-specific local state and uses structured schemas with rule-based patch operations to update memory instead of free-form summaries. Experiments on the ALFWorld benchmark show that this schema-guided memory design can substantially improve success rates and reduce invalid actions compared to several existing agent frameworks

**Compliance With Llm Reviewing Policy:**

Affirmed.

**Final Justification:**

The paper is clearly structured, well-motivated, and presents strong empirical results on the chosen benchmark. The proposed approach is conceptually clean, and the design of hierarchical, schema-governed memory is both intuitive and practically appealing.

Overall, the authors provided substantive and thoughtful clarifications addressing the weaknesses I identified in the paper. In particular, the additional experiments convincingly demonstrate robustness, and the ablation analysis clearly shows that the improvements are not primarily driven by action pruning as a potential confound. These additions significantly strengthen the soundness of the empirical evaluation and increase confidence in the reported results.

At the same time, my main concern remains only partially addressed. Specifically, the question of generalization (how well the proposed schema-based approach extends beyond the ALFWorld benchmark) remains open. While the authors provide hypothesis about applicability to broader, more open-ended environments (e.g., WebArena or SciWorld), this claim is not supported by direct empirical evidence and therefore remains somewhat speculative. Given that this concern was also raised by multiple reviewers, it appears to be a central limitation of the current work.

From the perspective of significance, the paper introduces an interesting and potentially impactful direction, especially for procedure-centric agent settings. However, the current evaluation suggests that the method is most effective in relatively structured environments with predefined task patterns, and it is not yet clear how well it scales to more diverse or less structured scenarios.

Taking everything into account, the rebuttal has strengthened my confidence in the technical soundness and robustness of the method but has not substantially changed my view regarding its scope of applicability. As a result, my overall assessment remains consistent with my initial evaluation.

Given the strengths of the paper and the quality of the rebuttal, I think the work meets the bar for acceptance. However, due to the limited diversity of evaluated scenarios and the open question of generalization, I find it reasonable to assign a score 4 (weak accept).

**Key Questions For Authors:**

1. **Generality beyond ALFWorld**
   Have you evaluated the proposed approach on more complex or less structured agent benchmarks (e.g., WebArena, SciWorld, or other tool-use environments)? Since ALFWorld contains a limited set of task types with largely fixed workflows, it would be interesting to understand how well the method scales to environments with more open-ended task structures

2. **Automatic or adaptive schema construction**
   In the current setup, the schema appears to rely on prior knowledge of the action space and task structure. Have you explored approaches for constructing these schemas automatically or adapting them during interaction, rather than defining them based on a prior knowledge of the environment?

**Limitations:**

The authors explicitly discuss the limitations of the proposed approach in the conclusion section, including its dependence on predefined schemas and task structures, as well as potential challenges in more open-ended environments

**Strengths And Weaknesses:**

### Strengths

- **Clear architectural formulation of hierarchical working memory**
  The paper proposes a structured framework that separates global task-level memory from subtask-specific local memory. This design provides a principled way to maintain decision-relevant information while avoiding uncontrolled growth of the interaction history

- **Schema-governed memory updates looks promising for SLMs**
  The agent’s memory is updated through structured schemas and simple rule-based patch operations. Instead of rewriting free-form textual summaries at every step, the model modifies specific fields in a controlled way. This makes the memory easier to maintain and reduces the risk of inconsistencies or information loss during long interactions. As noted by the authors in the conclusion, small language models often fail due to poor contextual information efficiency, and the proposed structured memory mechanism represents a promising direction for addressing this limitation

- **Training-free design with strong empirical results on an agent benchmark**
  The method avoids additional model training and instead relies on structured state updates and predefined schemas, making it particularly appealing for small language models. The experiments on ALFWorld demonstrate significant improvements in success rate and reductions in invalid action rates compared to several baseline methods

### Weaknesses

- **Dependence on predefined task schemas and workflows**
  The framework relies on an offline schema induction step that specifies the sequence of subtasks, the structure of the memory, and the admissible actions. While this enables strong performance within a task family, it may limit the generality of the approach in settings where task structures are less predictable or require dynamic replanning

- **Potential confounding effect of action space pruning**
  The method introduces an “effective action set” for each subtask, restricting the actions that the agent may generate. While this improves reliability, it also changes the decision space compared to several baselines. The paper does not provide ablations isolating the contribution of action pruning from that of the hierarchical memory architecture itself

- **Limited experimental validation and robustness analysis**
  The evaluation is conducted primarily on a single benchmark environment (ALFWorld), which contains a limited set of task types and largely fixed workflows. In such settings, part of the planning structure may effectively be specified in advance through the manually designed schema and subtask decomposition. As a result, it remains unclear how well the proposed approach would scale to environments with less predictable task structures and more open-ended interaction spaces (for example WebArena, SciWorld, tool-use benchmarks e.t.c.) In addition, the paper does not report results across multiple random seeds, making it difficult to assess the robustness of the reported improvements

- **Limited coverage of related work on hierarchical reasoning and planning**
  The discussion of related work could be broadened, particularly with respect to recent approaches that introduce hierarchical structures for reasoning and planning in LLM systems. For example, the paper does not discuss *HyperTree Planning: Enhancing LLM Reasoning via Hierarchical Thinking* (Gui, R., Wang, Z., Wang, J., Ma, C., Zhen, H., Yuan, M., ... & Wu, F. (2025). Hypertree planning: Enhancing llm reasoning via hierarchical thinking. arXiv preprint arXiv:2505.02322. and https://proceedings.mlr.press/v267/gui25b.html) , which also explores hierarchical decomposition to improve long-horizon decision making. More generally, a broader comparison with hierarchical planning and reasoning frameworks would help better position the proposed hierarchical memory architecture within the growing literature on structured agent reasoning

---

> ### Author Rebuttal · Authors · 2026-03-31
>
> # Generalization under Predefined Schemas (W1, W3, Q1)
>
> Our framework targets procedure-centric agent tasks, a primary deployment scenario for SLMs. Characterized by a shared macro-level planning structure and signalable subtask completion, these tasks capture a large and practically significant class of agent workloads. This aligns with recent Nvidia perspectives [1], which position the repeated execution of specialized procedures as the core value proposition for SLM deployment.
>
>  For more open-ended environments such as WebArena or SciWorld, the planning layer would require richer structures, but the schema-governed local memory mechanism remains directly applicable as a general interface for stabilizing execution under limited context.
>
> ---
>
> # Action Pruning as a Potential Confound (W2)
>
> We conducted component-level ablations on all 145 valid_unseen tasks (Qwen3-4B FP16) to isolate the contribution of each design choice. Our framework comprises three components: ① hierarchical goal-conditioned memory representation, ② action templates, and ③ patch constraints.
>
> The primary gain comes from the hierarchical memory representation, not action pruning. Component ① alone—without any action templates or patch constraints—achieves 69.7% success, already more than double MemGPT (31.7%).
>  ② and ③ each address specific SLM failure modes: patch constraints add +14.4pp by preventing autoregressive copy-corruption during full JSON rewrites, while action templates add +8.9pp by suppressing irrelevant action patterns that distract SLMs under limited context. Together they bring the full system to 91.0%. Detailed ablation results are provided in our response to Reviewer N6FM (Table therein) and will be included in the revised manuscript.
>
> ---
>
> # Relation to Hierarchical Planning (W4)
>
> Hierarchical planning is complementary to our contribution. Concretely, HyperTree Planning [2] constructs a hierarchical search tree at reasoning time to decompose complex problems into progressively refined sub-problems, improving the quality of the plan itself. Our work operates at a different layer: given a subtask decomposition, we maintain hierarchical working memory at execution time to ensure the SLM retains decision-relevant state across steps under tight context budgets. The two mechanisms are naturally composable, where hierarchical planning methods serve as an upstream planner to produce subtask structures, while our schema-governed memory stabilizes downstream execution.
>
> We will add the suggested citation and broaden the related-work discussion to explicitly position hierarchical planning as a complementary direction in the revised manuscript.
>
> ---
>
> # Experimental Robustness (W3)
>
> All reported experiments use greedy decoding (temperature = 0), ensuring reproducibility under a fixed protocol.
>
> To demonstrate robustness, we conducted additional experiments across multiple random seeds. With temperature = 0.7 and five random seeds, success rates are 0.937, 0.931, 0.917, 0.952, and 0.924, showing low variance and consistent performance.
>
> ---
>
> # Schema Construction and Adaptability (Q2)
>
> **Schema content is LLM-induced from traces, not manually designed from domain knowledge.**
> The offline pipeline consumes five randomly sampled successful trajectories per task family and proceeds in two LLM-driven stages. Stage 1 (Structure Extraction) analyzes the trajectories to identify the shared subtask sequence, per-subtask termination conditions mapped to observable state fields, and effective action patterns. Stage 2 (Schema Synthesis) takes Stage 1 outputs together with a base schema and generates a family-specific schema defining all local memory fields. The only manually designed artifact is the base schema for the simplest pick-and-place family, which scaffolds all subsequent induction.
>
> **Online adaptation is a natural extension of the current design.**
> The patch-based update mechanism and structured RETURN fields provide a principled interface for detecting schema violations or unexpected states. Recent work on agent workflow memory [3] demonstrates that streaming-based induction can effectively improve generalization. Similar mechanisms could incrementally refine schemas during deployment based on execution feedback.
>
> ---
>
> We hope these clarifications address the reviewer’s concerns. Code and induced schemas will be released upon publication.
>
> # Reference
>
> [1] Belcak et al., "Small Language Models are the Future of Agentic AI", arXiv 2025
>
> [2] Gui et al., "HyperTree Planning: Enhancing LLM Reasoning via Hierarchical Thinking", ICML 2025
>
> [3] Wang et al., "Agent Workflow Memory", ICML 2025

---

> > ### Author Rebuttal · Reviewer_9N4f · 2026-04-03
> >
> > Dear authors, thank you very much for your detailed responses!
> >
> > Overall, the authors provided substantive and thoughtful clarifications addressing the weaknesses I identified in the paper. In particular, the additional experiments convincingly demonstrate robustness, and the ablation analysis clearly shows that the improvements are not primarily driven by action pruning as a potential confound.
> >
> > However, the response to my main concern (the capabilities and generalization of the proposed schema beyond a single benchmark) remains less convincing. While the authors outline a hypothesis regarding applicability to broader settings, this claim is not supported by direct empirical evidence and remains largely speculative. This concern was not unique to my review and appears to be a broader issue raised by multiple reviewers.
> >
> > Given the overall strength of the work and the quality of the rebuttal, I believe it is reasonable to recommend acceptance. At the same time, due to the significant limitation in the diversity of evaluated scenarios, I do not find it appropriate to raise my score beyond 4 (weak accept), which is consistent with my initial assessment.

---

> > > ### Author Response · Authors · 2026-04-08
> > >
> > > We sincerely thank the reviewer for the positive assessment of our work and the quality of the rebuttal, as well as for the candid feedback throughout this process. The concerns raised in the first round regarding ablation and robustness have significantly deepened our understanding of the framework’s behavior.
> > >
> > > We fully agree that more open-ended environments such as WebArena or SciWorld would provide important additional evidence, and we are actively extending the framework in that direction. Due to rebuttal-time constraints, however, we cannot report complete cross-benchmark results here. We will clarify this scope more explicitly in the final manuscript and better highlight the robustness, ablation, and related-work evidence already reported in the paper.
> > >
> > > More broadly, the explicitly structured fields in local memory make the framework better suited to more complex control flow and demonstrate a degree of scalability.
> > > At the same time, more intricate planning structures remain a bottleneck for broader generalization, which motivates our ongoing efforts in this direction.

---

### Decision · Program_Chairs · 2026-04-30

**Decision:**

Accept (regular)

**Comment:**

This paper proposes a training-free hierarchical working memory framework for small language model agents, with goal-conditioned local memory, task-level global memory, schema-governed updates, and offline schema induction from a small number of successful traces. I recommend acceptance. The paper has a clear and practically meaningful contribution. The central idea, making decision-relevant state explicit for SLM agents instead of relying on free-form natural-language summaries, is well motivated, and the empirical gains on ALFWorld are substantial. In particular, the work shows that a relatively small model can achieve very strong long-horizon performance without finetuning or online LLM calls, which is an important result for low-cost agent deployment.

The rebuttal strengthened the paper in several important ways. Most notably, the additional ablations help separate the contribution of the hierarchical memory representation from action-space restriction, and the added robustness analysis increases confidence that the reported gains are not fragile. I also appreciate the clarifications around offline schema induction and the intended applicability boundary of the method. The main remaining limitation is generality. The current evaluation is still concentrated on ALFWorld, which is a relatively structured benchmark with reusable task-family regularities. As a result, it remains unclear how well the approach extends to more open-ended environments with branching, rollback, or less stable task structure. I think this concern is real, but it does not outweigh the current strengths. The paper makes a strong case for procedure-centric agent settings, even if broader claims should be stated more carefully.